# Personalised Esomeprazole and Ondansetron 3D Printing Formulations in Hospital Paediatric Environment: I-Pre-Formulation Studies

Mariana Ferreira [1], Carla M. Lopes [1,*], Hugo Gonçalves [2], João F. Pinto [3] and José Catita [1,2,*]

[1] Instituto de Investigação, Inovação e Desenvolvimento (FP-I3ID), Biomedical and Health Sciences Research Unit (FP-BHS), Faculdade de Ciências da Saúde, Universidade Fernando Pessoa, Rua Carlos da Maia 296, 4200-150 Porto, Portugal

[2] Paralab, SA, Rua Dr. Joaquim Manuel Costa 946 B, 4420-437 Valbom, Portugal

[3] iMed-Research Institute for Medicines, Faculdade de Farmácia, Universidade de Lisboa, Av. Prof. Gama Pinto, 1640-003 Lisboa, Portugal

* Correspondence: cmlopes@ufp.edu.pt (C.M.L.); jcatita@ufp.edu.pt (J.C.); Tel.: +351-225-074-630 (C.M.L. & J.C.)

**Abstract:** Individualised medicine demands the formulation of pharmacotherapy in accordance with the characteristics of each patient's health condition, and paediatrics is one of the areas that needs this approach. The 3D printing of oral doses is one method for achieving customised medicine in paediatrics. In this work, pre-formulation studies were conducted to evaluate the viability of using specific raw materials to produce 3D printed dosage forms based on two active pharmaceutical ingredients (APIs), ondansetron and esomeprazole, which are important for therapeutic customisation in paediatrics. Pre-formulation studies were carried out by characterising the physical and chemical properties of selected raw materials, selected APIs and their mixtures, using analytical methods such as scanning electron microscopy (SEM), X-ray powder diffraction (X-RPD), simultaneous thermal analysis (STA) and differential scanning calorimetry (DSC). The flowability of powders, compatibility and stability studies were also performed. Among all the ingredients selected, the PVPs (K17, K25 and K90) had the best characteristics to incorporate both forms of Esomeprazole Mg in a formulation to produce extrudates. The results obtained validated the use of some selected raw materials for tablet manufacture by the 3D printing approach.

**Keywords:** 3D printing; paediatric; esomeprazole; ondansetron; pre-formulation studies

## 1. Introduction

Paediatric patients are subpopulations with unique characteristics, distinct from patients of other age groups. In physiological terms, there are factors inherent to age that decisively determine the response to pharmacotherapy [1].

The dose for paediatric patient therapy is traditionally calculated by extrapolating the used dose for adults to the child. However, this practice has not always led to adequate results, as the differences between adults and children are not limited to the size and weight of the body [2]. In childhood, the functional maturity is distinct from that observed in adults and the elderly, with differences beyond the anatomy, namely physiological and biochemical, which affect the pharmacokinetics (i.e., the bioavailability of drugs) and the pharmacodynamics (i.e., the therapeutic effect) [3–5]. In addition, the psychological and behavioural components which make the paediatric patient more resistant to therapeutic compliance, often leading to medication rejection, should not be underestimated. This resistance may result from several factors, including the existence of a complex dosage regimen, the use of unpleasant routes of administration (e.g., intravenous administration and difficulty in swallowing), the palatability of formulation, or the difficulty to explain the relevance and need of therapy to children [6–8]. Considering the aforementioned

characteristics of pharmacotherapy in paediatrics, the need to develop customised medicine is paramount to adjust current available therapeutic protocols to the doses required for these patients.

Esomeprazole and ondansetron are two important APIs that required individualised paediatric therapies. Esomeprazole belongs to the class of proton-pump inhibitors (PPIs) which is indicated to treat various gastrointestinal pathological conditions [9]. The safety and tolerability of PPIs drugs in paediatrics is a focus of attention for health professionals, making it necessary to adapt the doses to the child being treated [10,11]. Ondansetron is an antiemetic used to prevent vomiting and nausea induced by chemotherapy, radiotherapy, and post-operative treatments. This drug is also used in the acute treatment of the cyclic vomiting syndrome [12]. Commercially available pharmaceutical preparations containing ondansetron are limited due to its low water solubility and not always the most appealing to paediatric population because of its unpleasant taste [13,14]. Despite ondansetron's high safety profile, this drug can cause serious side effects when combined with other therapeutic agents such as arrhythmia and dosage-dependent QT-interval elongation, which demands dose control and monitoring, mainly in vulnerable populations [15]. The individualisation of pharmacotherapies which use these APIs in paediatrics is important to respect pharmacokinetic differences observed in different age groups, mainly when used in association with other complex therapies, such as chemotherapy and radiotherapy [16].

Every day, caregivers around the world struggle with the fact that many medicines provided to children are not child-friendly or have the proper dosage, forcing them to overcome these barriers by adapting commercialised pharmaceutical forms (e.g., splitting or crushing tablets) on a daily basis [7]. Conventional pharmaceutical compounding strategies allow the adjustment of dose required in the most suitable pharmaceutical form, according to the characteristics of the patient, namely fractioning (i.e., divisibility of commercial pharmaceutical forms), gridding (i.e., reduction to powder of dry solid fragments resulting from the fractionation process) and preparation of liquid dosage forms (i.e., dissolving or dispersing a solid compound in a liquid vehicle, usually aqueous) [17]. However, these strategies are not appropriate to be applied in all individualisation scenarios [18–20].

The use of innovative techniques to produce pharmaceutical forms in the hospital environment, with precise and accurate doses individualised for each patient, is crucial. The three-dimensional (3D) printing technique fits into this context because it has a large potential for the manufacture of medicines, such as individualised medication (including the production of modified drug release systems), the preparation of systems with high flexibility on dosing (useful in clinical trials), the reduction in residues resulting from conventional therapeutic individualisation practices (environmentally friendly), and it is an alternative decentralised method for the manufacture of medicines in situations of inaccessibility to the conventional pharmaceutical distribution network [21].

Several 3D printing techniques are reported in the literature for the manufacture of an object with almost any required shape previously designed with a proper software [22]. The use of 3D printing technology in the pharmaceutical and healthcare sectors is a relatively new concept, although research and development in the latter is extensive. The introduction of 3D printing facilitates the development of cost-effective, user-friendly, and tailored dosage forms with the desired drug release profiles, according to the needs of individuals or specific patient groups [23]. The application of this technique in pharmacy results in a new approach to prepare oral drug dosage forms, and 3D printing technology has the potential to shift the emphasis from the "drug" to the "patient." In addition to finely adjusting drug dosage based on patient information, personalised drug therapy can provide tablets in the shape, taste, and colour preferred by the patient which is highly relevant in paediatric populations [24]. Binder jetting, material extrusion, material jetting, and powder bed fusion are the most common 3D printing technologies used in pharmacies. Fused deposition modelling (FDM) printing via material extrusion is currently the most widely used technology [24,25]. FDM is carried out by heating a filament made of a thermoplastic polymer based, at a temperature close to its melting point. The extruded

product is deposited layer by layer of varying thicknesses (i.e., resolution) depending on the printer's nozzle and instructions provided to the printer by the software [25].

Pre-formulation studies represent a critical phase in the development of new medicines, involving the analysis and evaluation of the characteristics of the APIs and excipients that may influence the formulation, the design, and the performance of the process [26]. Several physio-chemical properties should be identified early in the development process, specifically during the pre-formulation stage. Indeed, they usually determine the formulation strategy to use in order to succeed further in development. It is also critical to assess the compatibility of APIs and excipients [27]. Pre-formulation studies are relevant not only for characterising the selected materials and confirming their use in the manufacturing of medicines through 3D printing, but also as reference data for further proof of concept studies. In this context, these studies have been carried out to assess the feasibility of various raw materials commonly used in the production of pharmaceutical products in processing technologies, such as extrusion, for the preparation of medicines by 3D printing.

Studies were conducted on the physical and chemical properties of raw materials, complementing the spare information (certificates of analysis) provided by the manufacturers. Additionally, this work aimed to evaluate the compatibility between the mixtures of selected raw materials (i.e., APIs and excipients), evaluating and validating their potential to be applied in the manufacture of printed tablets. To the best of our knowledge, no studies on the compatibility between the selected APIs, specifically esomeprazole and ondansetron, and the chosen excipients have been reported in the literature and only one study on the possible manufacture of printed medicines containing ondansetron has been published [14]. Even though, it did not use the excipients chosen in the presented study nor the intended 3D production technique (FDM).

## 2. Materials and Methods

### 2.1. Materials

Esomeprazole magnesium trihydrate (USP, 99.5% pure and a water content of 7.32% (*w/w*), Everest Organics Ltd., Telanagana, India) was presented as a white to slightly coloured powder. Esomeprazole magnesium dihydrate (European Pharmacopoeia—Eur. Pharm., 99.6% pure and a water content of 5.3% (*w/w*) measured by Karl Fischer method, Minakem SAS, Beuvry, France) was presented as a white to slightly coloured powder. Ondansetron hydrochloride (USP, 105% pure and a water content of 10.1% (*w/w*), Shodhana Laboratories Ltd., Telangana, India) was presented as a white to slightly coloured powder. Microcrystalline cellulose (Avicel®, FMC Corp, Philadelphia, PA, USA), polyvinylpyrrolidone k17, k25 and k90 (PVP, Acofarma®, Terrassa, Spain) and polyvinyl alcohol (PVA Mw30, Acofarma®, Terrassa, Spain) were the selected excipients. These substances are classified as GRAS (generally recognised as safe), demonstrating consensus on their safe use by authorities. Additionally, the existence of published studies on the application of these substances in 3D printing production, namely fused deposition modelling (FDM) validates the selection of these materials in the studies carried out.

### 2.2. Methods

All ingredients were characterised in terms of moisture content, flowability, morphology, particle size distribution, physical structure of the solid state, and thermal behaviour. Subsequently, mixtures of the raw materials (APIs and excipient) were also thermally characterised.

#### 2.2.1. Loss on Drying

The moisture content of samples was determined by loss on drying using the MB35 Moisture Analyzer (Ohaus, Nänikon, Switzerland) at a temperature of 105 °C, according to European Pharmacopoeia 8.0. Samples weighing between 0.500 g and 0.600 g were uniformly distributed on aluminium supports. During the procedure, values of the initial mass, final mass after drying, and the respective percentage of the moisture content were

registered. The samples were analysed in triplicate, and the average and variability (SD) were determined.

### 2.2.2. Flowability—Tapped Density of Powders

The tap density of various powders was determined in accordance with the European Pharmacopoeia 8.0 method (European Medicine Agency, 2016). A sample of each raw material was weighed in a measuring cylinder and the powder volume was registered. The graduated measuring cylinder was secured on a mechanical tapping machine (D-63 Heusenstamm by Erweka®, Langen, Germany) and subjected to the mechanical effect of 10, 100, 500 and 1250 taps. At the end of each tapping series, the volume occupied by the powders (V10, V100, V500 and V1250) was registered. When the difference between V500 and V1250 exceeded 2 cm$^3$, a further 1250 taps were required. The determination of the tap density of powders test was then used to calculate the compressibility index (CInd), the Carr index (CI) and the Hausner ratio (HR). The CInd, calculated by the equation CInd = ((V0 − V500)/V0) × 100), describes a property that influences the flow characteristics of a powder [23]. In addition, there are other equations that evaluate the capacity of a powder to reduce its volume through the effect of mechanical beats, such as the Carr index (CI) and the Hausner ratio (HR), which are represented by the respective equations:

$$CI = ((d500 - d0)/d500) \times 100$$

$$HR = d500/d10$$

where d0 corresponds to the apparent density before compaction, d10 and d500 correspond to the density after 10 and 500 taps, respectively [23].

The samples were analysed in triplicate, and the average and variability (SD) were calculated.

### 2.2.3. Scanning Electron Microscopy (SEM)

SEM images were collected by a Phenom ProX (ThermoFisher Scientific®, Bleiswijk, The Netherlands) enabling the analysis of both the granulometry and the morphology of raw materials. Voltages between 5 and 15 kV were applied in the analysis of the raw materials. The images were obtained using the Automated Image Mapping software (version 1.92.0) and data on particle size were acquired using the Particle Metric software (version 2.0) (PhenomWorld—ThermoFisher Scientific®, Bleiswijk, The Netherlands). A minimum of 7000 particles were measured per sample.

### 2.2.4. X-ray Powder Diffraction (X-RPD)

The crystallinity of each material was evaluated by X-RPD technique using MiniFlex 600 apparatus from Rigaku® (Tokyo, Japan). MiniFlex 600 features an x-ray generator with copper anode and maximum power of 600 W, a goniometer with Bragg–Brentano geometry (typology θ/2θ), with angular amplitude −3 to +145° and accuracy ±0.02°. Measurements were obtained using a zero-circular background with silicone support and performed at an analysis rate of 5°/min from 3 to 55°.

### 2.2.5. Simultaneous Thermal Analysis (STA)

STA is an analytical technique combining differential scanning calorimetry (DSC) and thermogravimetric analysis (TGA). Both DSC and TGA techniques were performed in the same real time and the sample was subjected to the same experimental conditions. STA analysis was applied to APIs and excipients samples independently. Simultaneous thermal analysis was conducted using STA 449 F3 Jupiter® (Netzsch®, Waldkraiburg, Germany). The analysis was performed under the following conditions: (i) exposure of the samples to a nitrogen atmosphere, (ii) temperature variation in a range from ambient temperature to 600 °C (heating rate of 10 °C/min) and (iii) analysis of the raw data with the NETZSCH® Proteus—Thermal Analysis v6.1 software (Waldkraiburg, Germany). The analysis of the physical mixtures of the raw materials was carried out only by DSC using

a DSC 214 Polyma equipment (Netzsch®, Waldkraiburg, Germany). Thermograms of the mixtures were obtained in a temperature range between 25 °C and 180 °C, applying a heating rate of 10 °C/min. Samples were exposed to a nitrogen atmosphere and two heating/cooling cycles were carried out. Before each cycle, the samples were stabilised to a temperature of 25 °C for analysis standardisation. In parallel, an empty aluminium crucible was used as a reference for the analysis. The data obtained from the tests were analysed by NETZSCH® Proteus—Thermal Analysis v7.1 software (Waldkraiburg, Germany).

## 3. Results and Discussion

### 3.1. Moisture Content and Flowability

All raw materials have been analysed for their moisture content and flowability (Table 1). Moisture content corresponds to the amount of non-essential water contained in a dry substance. This determination is important to be carried out on components for pharmaceutical use, since water can influence the resulting products at various levels, namely the mixture homogeneity, and stability of the resulting solid pharmaceutical form.

**Table 1.** Results obtained for the moisture content (loss on drying and thermal analysis) and flowability properties.

|  | Moisture Content (%) | TGA 1st Mass Change (%) | CInd (%) | CI (%) | HR |
|---|---|---|---|---|---|
| MCC | $8.00 \pm 0.16$ | <2.00 | $22.0 \pm 1.3$ | $21.8 \pm 1.3$ | $1.18 \pm 0.03$ |
| PVP K17 | $14.20 \pm 0.28$ | 12.36 | $16.3 \pm 1.0$ | $16.5 \pm 1.2$ | $1.15 \pm 0.03$ |
| PVP K25 | $15.70 \pm 0.31$ | 16.78 | $21.0 \pm 1.2$ | $21.0 \pm 1.3$ | $1.17 \pm 0.02$ |
| PVP K90 | $13.83 \pm 0.22$ | 12.25 | $17.0 \pm 1.0$ | $17.1 \pm 1.2$ | $1.09 \pm 0.02$ |
| PVA Mw30 | $5.30 \pm 0.12$ | 4.87 | $17.7 \pm 1.1$ | $17.2 \pm 1.2$ | $1.11 \pm 0.03$ |
| Esomeprazole Mg ($2H_2O$) | $2.58 \pm 0.10$ | 5.19 | $23.4 \pm 1.6$ | $24.3 \pm 1.3$ | $1.26 \pm 0.04$ |
| Esomeprazole Mg ($3H_2O$) | $2.97 \pm 0.09$ | 7.73 | $20.0 \pm 1.5$ | $20.1 \pm 1.3$ | $1.21 \pm 0.03$ |
| Ondansetron | $10.29 \pm 0.22$ | 9.59 | $17.0 \pm 1.4$ | $17.0 \pm 1.2$ | $1.11 \pm 0.04$ |

Abbreviations: CI (%)—Carr index; CInd (%)—compressibility index; esomeprazole Mg ($2H_2O$)—esomeprazole magnesium dihydrate; esomeprazole Mg ($3H_2O$)—esomeprazole magnesium trihydrate; HR—Hausner ratio; MCC—microcrystalline cellulose; PVA—polyvinyl alcohol; PVP—polyvinylpyrrolidone; TGA—thermogravimetric analysis.

Although there is no maximum moisture content limit for solid ingredients (powders), it is desirable that it be less than 8%, in order to minimise particle agglomeration and other problems during the manufacturing of solid dosage forms [28]. An inadequate moisture content of the powders to be mixed may result in problems of homogeneity of the mixture and instability of the resulting solid dosage form. Powders with an excessive moisture content have particles that agglomerate more easily, resulting in changes in the dissolution profile and forming non-resistant solid dosage forms (i.e., more friable, and less resistant to rupture). These impacts may result in changes in the technological ability and stability during the storage period of the pharmaceutical product [29]. The studied components' moisture contents were shown to range from 2.58 to 15.70%. In the case of both esomeprazole Mg, these values ($2.58 \pm 0.1\%$ and $2.97 \pm 0.09\%$) indicated the contribution of adsorbed and hydration water, and the wetting water in the other ingredients. Results obtained by loss on drying differed from those obtained by thermal analysis and reported in the manufacturer's analytical certificate in the cases of MCC and both forms of esomeprazole Mg. However, these results are in good agreement with the first mass change determined by thermal analysis. In the case of esomeprazole, these differences are due to the temperature used in the loss on drying method (105 °C), which is lower than the dehydration temperatures (approximately 130 °C to 160 °C). Sun [30] noted that differences are explained by the atmosphere in which the analysis was performed. In the case of loss on drying, analysis was performed at a normal environment (relative humidity over 60%). On the other hand, thermal analysis were performed under nitrogen, being the sample stabilised for a few minutes before starting the temperature program.

The assessment of solid material flowability is an important step since poorly flowing powders may be disadvantageous for the formulation production, leading to loss of material by adhesion to equipment surfaces and less homogeneous extrudates [31]. The raw materials analysed presented intermediate values when compared to their reference values [32,33]. In fact, the values for CInd and CI% ranged between 16.3 and 24.3%, and the values for HR, ranged between 1.09 and 1.26. Thus, the raw materials exhibited satisfactory flow properties, being esomeprazole Mg $2H_2O$ the raw material with the highest CInd and HR values, and, therefore, having the worst flow characteristics. PVP K17 and K90 were the raw materials with the best flow properties.

### 3.2. Morphology and Particle Size Distribution

The particle size distribution and morphology of all materials were measured using SEM. Particle sizes of the raw materials ranged from 6.7 and 143 nm. In terms of morphology, the particles presented irregular and angular shapes in the case of MCC; PVP K90; PVA Mw30 and ondansetron. For PVP K17; PVP K25; esomeprazole Mg $2H_2O$ and esomeprazole Mg $3H_2O$ particles were round (Figure 1).

MCC was the excipient with the smallest particles, as opposed to PVP K90 (Table 2). When comparing the two forms of esomeprazole, the dihydrate form exhibited smaller particle sizes than the trihydrate form. This effect can be an advantage since smaller particle sizes have a greater surface area and dissolution rate, resulting in better oral bioavailability. Additionally, smaller particles can be better involved by the polymer in the extrudate, leading to a higher dissolution within the polymer matrix.

**Table 2.** Particle size analysis of raw materials by scanning electron microscopy (distribution by number).

| Property | Circle Equivalent Diameter (µm) | | | |
|---|---|---|---|---|
| | Average | D10 | D50 | D90 |
| MCC | 41.2 | 17.8 | 38.6 | 66.8 |
| PVP K17 | 66.0 | 26.6 | 64.2 | 104.0 |
| PVP K25 | 71.3 | 23.6 | 68.4 | 116.0 |
| PVP K90 | 143.0 | 63.7 | 142.0 | 224.0 |
| PVA Mw30 | 139.0 | 53.6 | 139.0 | 217.0 |
| Esomeprazole Mg ($2H_2O$) | 6.7 | 2.7 | 6.1 | 12.0 |
| Esomeprazole Mg ($3H_2O$) | 19.8 | 6.4 | 17.1 | 36.9 |
| Ondansetron | 114.0 | 41.0 | 101.0 | 208.0 |

### 3.3. Crystallinity

The analysis of the crystalline structures of the various raw materials was evaluated by XRD. According to the peak's characteristics obtained in the diffractograms, it was possible to verify whether each material presented a predominant crystalline or amorphous physical structure. Figure 2 illustrates two diffractograms of different solid structures—crystalline (ondansetron) and amorphous (PVP K17). The XRD diffractograms of the other raw materials are presented in the Supplementary Materials (Supplementary Figures S1, S3, S5 and S7).

In the Figure 2 diffractogram a, the peaks were short and very broad. This demonstrates the recording of low intensity refractory beams, indicating the existence of a predominantly amorphous solid structure. This type of structure was observed in the diffractograms of PVP K17, PVP K25, PVP K90 and PVA Mw30. The diffractogram b of the Figure 2, corresponding to the ondansetron, presented well-defined peaks. The presence of well-represented diffraction maxima (peaks), with narrow and high symmetry, indicates the existence of a crystalline solid structure.

The three APIs presented mainly a crystalline solid structure, although in different degrees (ondansetron higher than 99%; esomeprazole Mg ($2H_2O$) approximately 82% and esomeprazole Mg ($3H_2O$) around 72%). The MCC exhibited a diffractogram compatible with a mixed structure composed by crystalline and amorphous phases (Supplementary Figure S1). These results are consistent with other ones reported in the literature [34–36]. In the study

reported by Browne et al. [34], the excipient PVA was found to have a partial crystalline structure. Ingredients with amorphous solid structures can be advantageous given that amorphous solids have a higher dissolution rate [37].

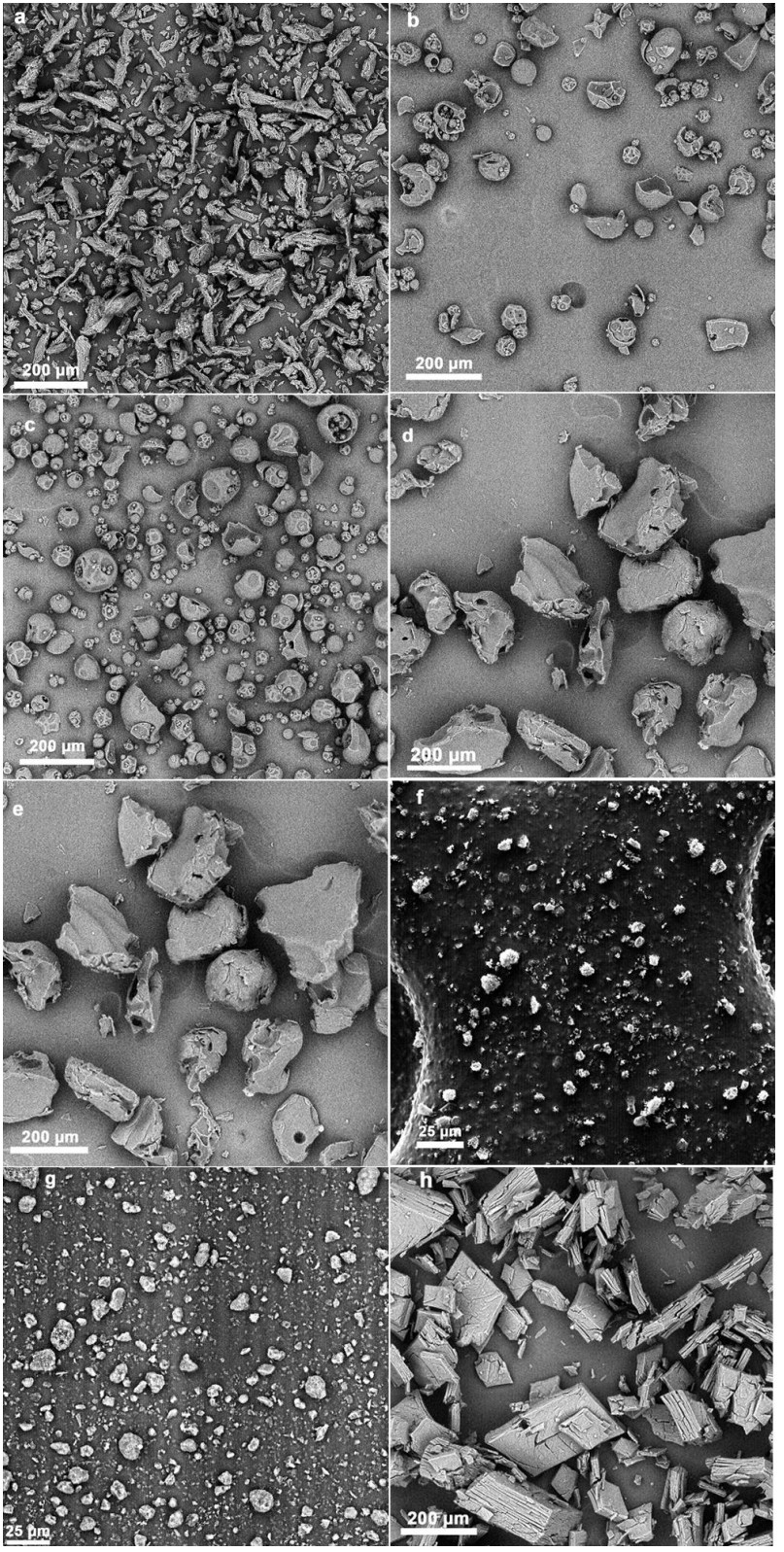

**Figure 1.** SEM images of particles (**a**) MCC; (**b**) PVP K17; (**c**) PVP K25; (**d**) PVP K90; (**e**) PVA Mw30; (**f**) Esomeprazole Mg ($2H_2O$); (**g**) Esomeprazole Mg ($3H_2O$); (**h**) Ondansetron.

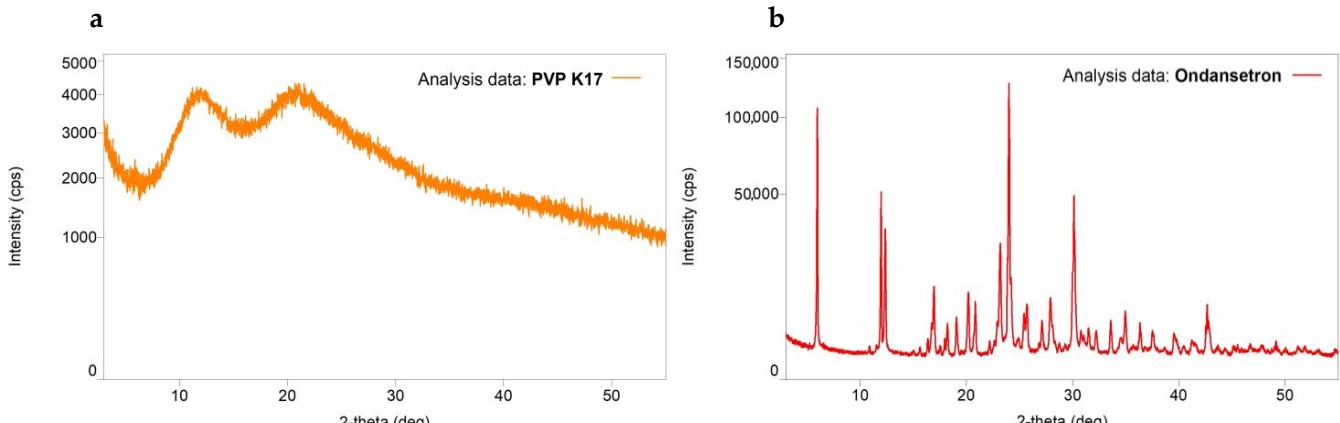

**Figure 2.** X-RPD diffractograms of PVP K17 (**a**) and Ondansetron (**b**).

### 3.4. Stability and Thermal Profile

The thermal behaviour of the raw materials was analysed using DSC and TGA, carried out simultaneously (STA) and alone (DSC).

The APIs were the raw materials with the lower thermal stability (other raw materials thermograms in the Supplementary Materials). STA thermograms of esomeprazole Mg $2H_2O$ and $3H_2O$ (Figure 3a,b) showed a small loss in sample mass of 5.19% and 7.73%, respectively, immediately after the heating started. These effects resulted from water loss by evaporation. In addition, at approximately 180 °C, a thermic phenomenon occurred in both samples, indicating a physical and/or a chemical transformation at that temperature. These recorded phenomena correspond to a fusion (endothermic peak), maximum at temperatures ranging from 149 °C to 165 °C, followed by a decomposition of the material (exothermic peak). In this sense, the two forms of hydration of esomeprazole Mg have shown thermal stability up to a temperature of around 180 °C. The STA thermogram of ondansetron (Figure 3c) presented an initial endothermic phenomenon, which was also observed in the analysis of the other raw materials, with a mass loss due to the evaporation of water. At a temperature of around 180 °C, an endothermic phenomenon compatible with fusion was noted, followed by endothermic and exothermic phenomena associated with a mass loss of 72.02%, compatible with material decomposition, indicating that ondansetron was stable up to 220 °C. The STA thermograms of other raw materials are presented in the Supplementary Materials (Supplementary Figures S2, S4, S6 and S8).

The STA thermogram of the MCC showed an endothermic phenomenon (heat absorption) that occurred at approximately 300 °C together with a mass loss of 82.65%, suggesting that the material was stable up to this temperature. The different PVP materials demonstrated stability up to a temperature of 400 °C. Below this temperature, samples did not show any thermal phenomena compatible with physical or chemical changes of the material. Concerning PVA Mw30, there was a pronounced mass loss (89.26%) at temperatures above 300 °C. The onset of the endothermic phenomenon at this temperature corresponds to the loss of thermal stability of the raw material, suggesting the degradation of the sample.

Table 3 displays the melting temperatures (measured at peak temperature) and thermal stability of raw materials. In summary, the melting temperature for crystalline or partially crystalline ingredients ranged from 149 °C (esomeprazole Mg $3H_2O$) and 338 °C (MCC), and the decomposition temperature ranged from 180 °C (esomeprazole Mg $3H_2O$) and 400 °C (all PVPs).

The thermal behaviour of esomeprazole Mg by the DSC was depicted in the study by Kumar et al. [38], in which the established melting temperature of esomeprazole magnesium was 175 °C. The material used in this study (both di and tri hydrate) presented lower melting temperatures. As stated before, crystallinity (as determined by XRD) was around 75%, resulting in a lower melting temperature and a wider melting interval. In fact, when compared to ondansetron (99% crystalline), esomeprazole presented wider melting

peaks. In the study carried out by Parhi and Panchamukhi [39], it is reported the thermal analysis of ondansetron, evidencing a phenomenon corresponding to melting with a peak at 183 °C, is compatible with the one observed in this study. The thermal stability of MCC was evaluated by others [40] that reported a decomposition temperature of approximately 300 °C, compatible with the one obtained in this study. The PVP thermal stability evaluation was also studied by others, such as in [41], where the temperature of decomposition measured was 400 °C, which is consistent with the values obtained here.

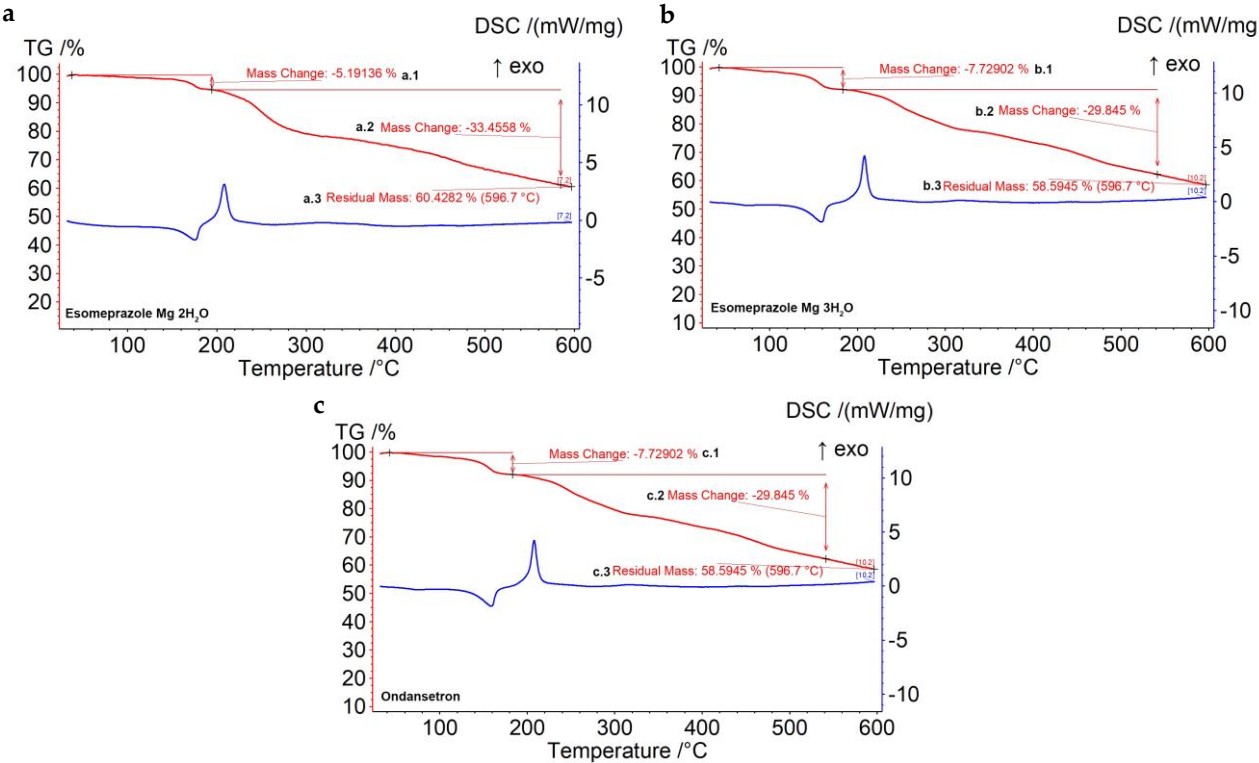

**Figure 3.** STA thermograms of active pharmaceutical ingredients (APIs): (**a**) Esomeprazole Mg $2H_2O$ (a.1—Mass Change: $-5.19\%$; a.2—Mass change: $-33.46\%$; a.3—Residual Mass: $60.43\%$), (**b**) Esomeprazole Mg $3H_2O$ (b.1—Mass Change: $-7.73\%$; b.2—Mass change $-29.83\%$; b.3—Residual Mass: $58.59\%$) and (**c**) Ondansetron (c.1—Mass Change: $-9.59\%$; c.2—Mass change: $-72.02\%$; c.3—Residual Mass: $17.94\%$).

**Table 3.** Melting temperature and thermal stability of raw materials as determined by STA analysis.

|  | Melting Point (T °C) [1] | Decomposition (T °C) |
|---|---|---|
| MCC | 338 | 300 |
| PVP K17 | 72 | 400 |
| PVP K25 | 82 | 400 |
| PVP K90 | 73 | 400 |
| PVA Mw30 | 180 | 280 |
| Esomeprazole Mg ($2H_2O$) | 164 | 200 |
| Esomeprazole Mg ($3H_2O$) | 149 | 180 |
| Ondansetron | 181 | 220 |

[1] Peak temperature of endothermic effect.

### 3.5. Compatibility between Raw Materials

The DSC analysis of physical mixtures of the selected raw materials was performed to assess the compatibility between them. The physical mixtures of raw materials were obtained by combining each excipient with an API (50:50 ratio) making a total of 15 mixtures. Figure 4 shows three DSC thermograms obtained for the physical mixtures of MCC with

each API. The assessment of their compatibility was made by comparing thermograms of each material with the ones obtained for physical mixtures. When mixture show a similar thermal behaviour to the ones observed when ingredients are analysed individually, this suggests their compatibility.

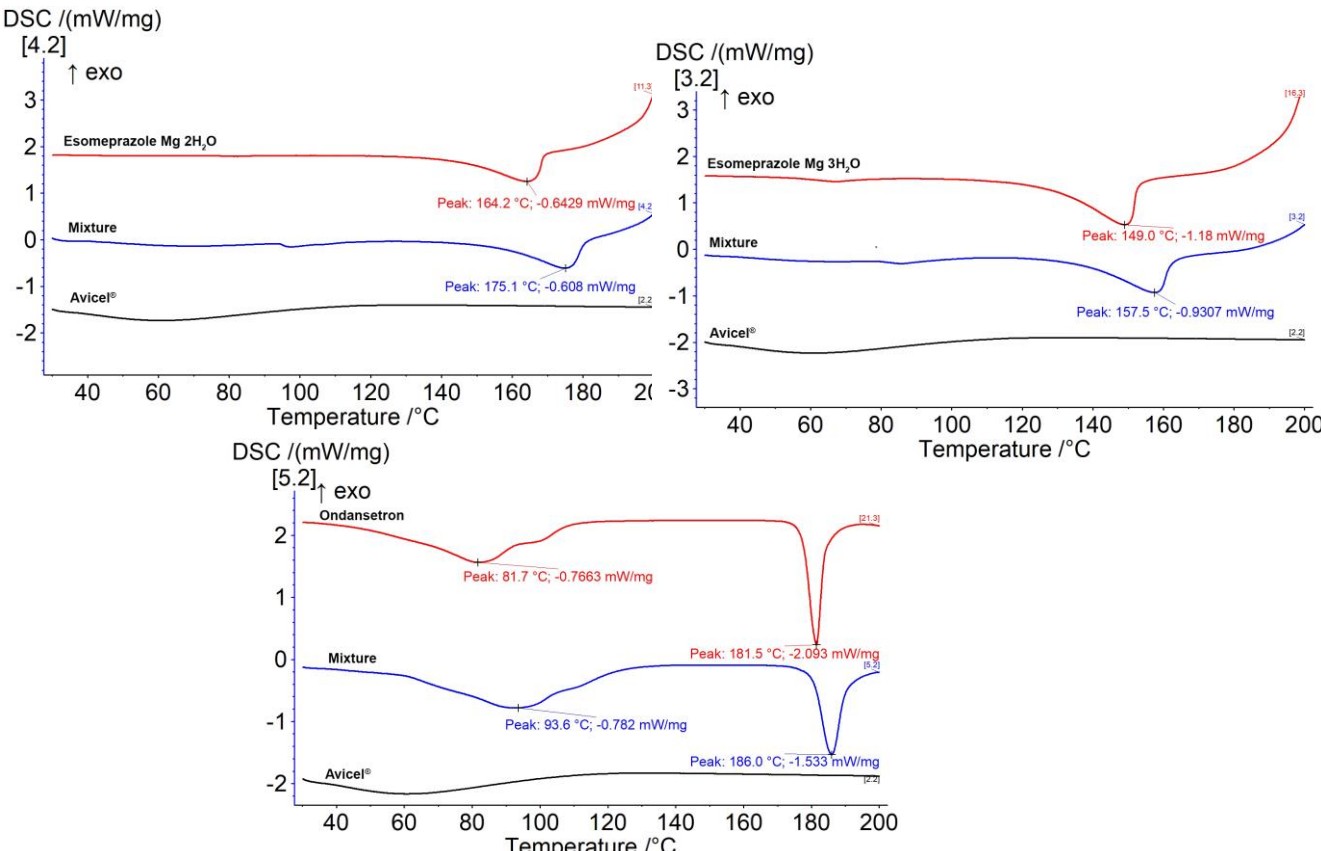

**Figure 4.** DSC thermograms of single materials (MCC, esomeprazole Mg $2H_2O$, esomeprazole Mg $3H_2O$ and ondansetron) and physical mixtures of MCC and APIs (50:50).

In the case of MCC analysis with each API, the obtained thermogram represents mainly the addition of the ones obtained for each individual ingredient. This means that there is no relevant chemical or physical interaction between these ingredients up to the studied temperature (i.e., 200 °C) (Figure 4). Considering the mixtures obtained with PVPs (K17, K25 and K90) and both esomeprazole Mg $2H_2O$, $3H_2O$, there were no notable changes in the thermograms, indicating no significant physical and/or chemical interactions in the mixtures up to the studied temperature (200 °C) (Supplementary Materials—Figures S9–S11). However, in the case of ondansetron, the mixture thermograms were clearly not additive in relation to the ones obtained for each individual ingredient (Figure 5).

For all PVPs studied, there was a temperature deviation for the first endothermic effect and a temperature deviation along with a peak widening in the second endothermic effect. These results suggested some extent of incompatibility between ondansetron and PVPs. Incompatibility of APIs with PVPs was also previously reported by Tita et al., for ketoprofen and PVP K-30 [42]. Finally, the DSC thermograms of the individual ingredients were compared with the thermograms of the PVA Mw30 mixtures to further evaluate the compatibility between them. An additive effect of the peaks was observed, suggesting that there was no noticeable incompatibility between materials. The DSC thermograms of all mixtures revealed an additive effect of the peaks in relation to the single raw materials (Supplementary Materials—Figures S9–S11).

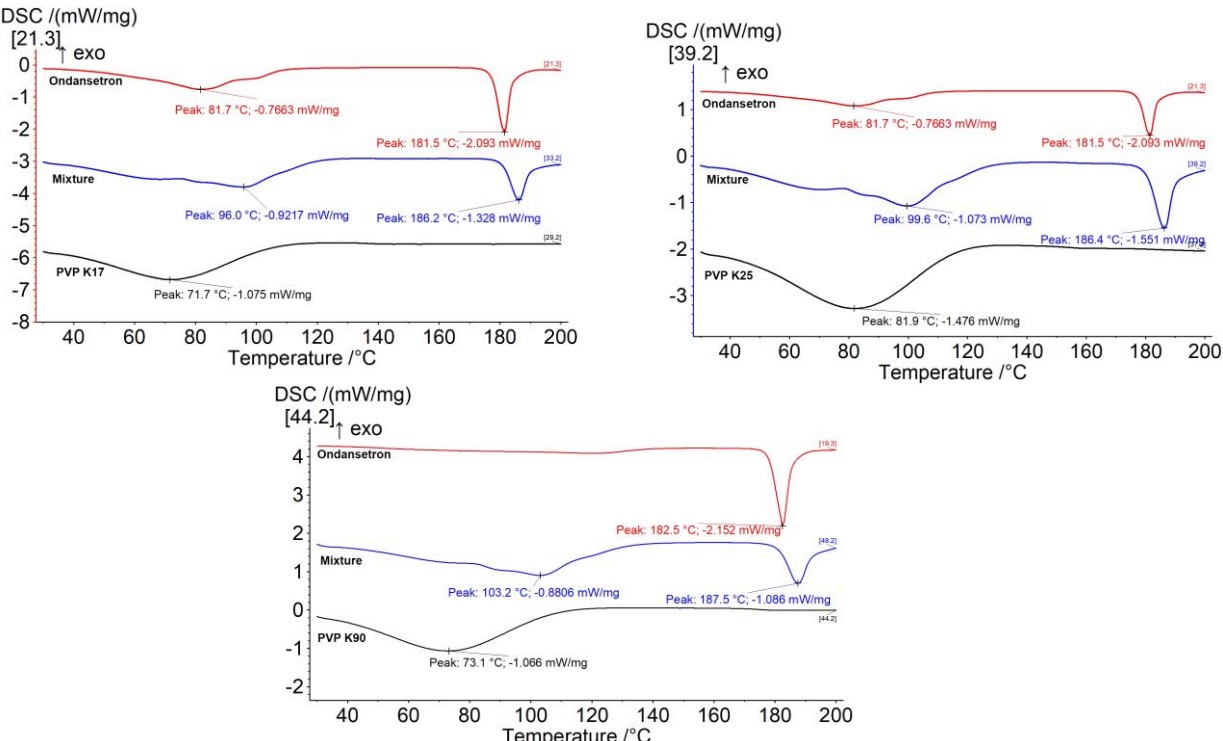

**Figure 5.** DSC thermograms of single materials and physical mixtures between ondansetron and PVPs (K17, K25 and K90).

## 4. Conclusions

Several ingredients (active and non-active) were characterised in terms of moisture content, flowability, particle size, morphology and X-RPD pattern. Its compatibility and stability were studied by the thermal methods of analysis, and X-ray diffraction patterns.

The values of moisture contents obtained using both techniques (i.e., loss on drying and thermal analysis) agreed except for MCC, the differences of which were related to environmental analysis conditions.

In the case of some binary mixtures, changes in the profile of thermograms (STA and DSC) indicate the production of some interactions as a function of heating. That was the case of mixtures of PVPs with ondansetron. For the other mixtures studied, no interactions were observed up to 200 °C.

Among all the ingredients selected, the PVPs (K17, K25 and K90) have the best characteristics to incorporate both forms of esomeprazole Mg in a formulation to produce extrudates. These excipients have a high thermal stability (up to 400 °C), and they begin to soften below 100 °C, which may be advantageous because the extrusion process should be carried out below 180 °C to ensure the thermal stability of the APIs. At the required temperature for the extrusion process, the PVPs are expected to produce a paste-like phase that allows the homogeneous incorporation of the APIs and an adequate consistency of the extruded product extrusion. For ondansetron, due to the possible incompatibility observed in the thermal analysis of physical mixtures with PVPs, PVA Mw 30 could be an alternative to produce extrudates. However, further studies should be conducted, or other ingredients such as some partial cross-linked hydrogels (e.g., hyaluronic acid or its derivates, Pluronics®, Buenos Aires, Argentina) may be considered to produce 3D-printed oral formulations.

**Supplementary Materials:** The following supporting information can be downloaded at: https://www.mdpi.com/article/10.3390/app122010585/s1, Figure S1: Diffractogram of microcrystalline cellulose (MCC) analysis by X-RPD technique, Figure S2: Microcrystalline cellulose (MCC) TG/DSC (STA) thermogram; Figure S3: Diffractograms of PVP K25 (a) and PVP K90 (b) analysis by X-RPD technique; Figure S4: PVP K17 (a), PVP K25 (b) and PVP K90 (c) TG/DSC (STA) thermograms; Figure S5. Diffractogram of PVA Mw30 analysis by X-RPD technique; Figure S6: PVA Mw30 TG/DSC (STA) thermogram; Figure S7: Diffractograms of esomeprazole Mg $2H_2O$ and $3H_2O$ analysis by X-RPD technique; Figure S8: Ondansetron TG/DSC (STA) thermogram; Figure S9: DSC mixtures thermograms of PVP K17 with esomeprazole: esomeprazole Mg $2H_2O$ (a) and esomeprazole Mg $3H_2O$ (b); Figure S10 : DSC mixtures thermograms of PVP K25 with esomeprazole: esomeprazole Mg $2H_2O$ (a) and esomeprazole Mg $3H_2O$ (b); Figure S11: DSC mixtures thermograms of PVP K90 with esomeprazole: esomeprazole Mg $2H_2O$ (a) and esomeprazole Mg $3H_2O$ (b); Figure S12: DSC mixtures thermograms of PVA Mw30 with different APIs: esomeprazole Mg $2H_2O$ (a); esomeprazole Mg $3H_2O$ (b) and ondansetron (c).

**Author Contributions:** Conceptualisation, C.M.L. and J.C.; Investigation, M.F. and H.G.; Methodology, C.M.L. and J.C.; Project administration, C.M.L. and J.C.; Supervision, C.M.L. and J.C.; Writing—original draft, M.F.; Writing—review and editing, C.M.L., J.F.P. and J.C. All authors have read and agreed to the published version of the manuscript.

**Funding:** This research received no external funding.

**Institutional Review Board Statement:** Not applicable.

**Acknowledgments:** The authors would like to thank the suppliers of APIs, Minakem® (France), who provided esomeprazole magnesium dihydrate micronised EUR, and Zim Laboratories Limited (India), who provided both esomeprazole magnesium trihydrate USP and ondansetron hydrochloride USP.

**Conflicts of Interest:** The authors declare no conflict of interest.

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
