# Peer review of "Personalised Esomeprazole and Ondansetron 3D Printing Formulations in Hospital Paediatric Environment: I-Pre-Formulation Studies"

_applsci, doi:10.3390/app122010585_

Round 1

Reviewer 1 Report

Interesting manuscript addressing an important 'medical need' in improving formulations for pediatric population. However, the work represented here concerns only the characterization of the materials. Based on the title I would have expected that also some test filaments would have been produced/tested. It would have been better to change the title in order to avoid 'misleading' (title starts now with ' personalized formulations...) and put more emphasize on ' pre-formulation' or characterization

other remarks:

Introduction:

- a clear motivation/explanation to develop 3D formulations for the two chosen drugs is missing, this is important as for both drug substances formulations for the paediatric population are available (granules or syrup) and the authors should make clear why their study is necessary and what it adds to what is already available (e.g. is there evidence that current formulations are not always applicable and why etc)

line 42-43: palatability of formulations has been known as important factor for acceptance by children, this is not mentioned in the introduction (e.g. reference Matsui. 2007. PPDT 8: 55-60)

Materials and Methods:

line 127: are the used excipients mentioned in the ' STEP' database (Safety and Toxicity of Excipients for Paediatrics)?

Results

line 372: compatibility is only assessed based on DSC analysis, this indicates thermal influences, however gives no indication on other possible chemical incompatibilities. I would have expected an analysis of different mixtures/test filament.

figure 4: avicel is mentioned in legend, article speaks of MCC, please put same name throughout the article

Discussion: the results are summed up, however a discussion is often missing, for instance on whether amorphous or crystalline is desired for the further development

Reviewer 2 Report

This article "Personalized Esomeprazole and Ondansetron formulations in hospital paediatric environment: I-Pre-formulation studies for developing extruded filaments for 3D printing" seems to be the first part of a bigger study. It is composed of 16 pages, 3 Tables, 5 figures, 1 supplementary file and 37 réf biblio (may be improved). Editing of the english language  may be required to improve this publication and enable the reader to understand the aim and the results of this study. This paper only focus on physico-chemical characterization without going deeper in the process than this API is compatible with this excipient. The outcomes of this paper are not, in my point of view sufficient, for a publication. But if combine with formulation, 3D printing optimization, ... it will give greater scope to promote this research.

The authors will find below some comments to improve their paper. 

Abstract

·         Line 23 : have need to be replaced by had

·         Line 25: validate need to be replaced by validated

Introduction

·         Line 59/60: the sentence needs to be rephrased not very understable

Results and discussion

·         Line 201: “accompanying” isn’t the best word here

·         Table 1: as the values are presented as mean +/- SD, the authors need to indicate if they have done triplicate or whatever, that enable them to calculate SD

·         Line 213: “powders of ingredients” ?

·         Line 224: “1st” ned to be replace by “first”

·         Line 227: (by other” need to be replace by the name of the first author et al.,

·         Line 229: how the authors controlled the relative humidity ?

·         Line 230: except if the authors performed only 1 thermal analysis, was need to be replaced by were

·         Line 42: for the size distribution part, why the authors didn’t used a laser diffractometer? It would have been more accurate (mean +/- sd) and more representative of the populations.

·         Line 269: not only figure S1 but also S3, S5, S7, … The authors need to update this part

·         Line 293: “low” and “wide” are not appropriate here. Wide could be replaced by “broad” for example

·         Line 296: the authors talks about diffractogram B but on the figure the diffractograms aren’t identified as it

·         Line 312: “the least thermal stability” ? Rephrase required ?

·         Line 328: Not only Figure S2. The authors need to update it

·         Figure 3/4/5 : Quality need to be improved

·         The authors need to be careful on the verb tense. They need to choose between past and present (usually it’s past since what is described was already done). The author mixed both and it’s not correct.

Conclusion

·         Seems to be a mix between discussion and conclusion. Discussion that may have been useful in the “results & discussion” part. The authors need to modify it

·         Why hydrogels could be interesting to produce extrudates especially since this process require to heat. It could be a drawback

Supplementary materials

·         Captions need to be corrected with the right numbering

Acknowledgments

·         Dunkerque isn’t a country but a city. The author need to correct it. Moreover, Minakem SAS isn’t located in Dunkerque but in Beuvry la Forêt, France.

References

·         Be careful for 9. Some formatting is required

Supplementary

·         14 figures, but in fact 15

·         Figure S1: Bigger chart legends are required

·         Figure S3: The 3 diffractograms need to be numbered (a/b/c or 1/2/3) and it need to be indicated in the caption

·         Page4: it’s not “Figure 9” but “figure S4. Graph quality need to be improved and as for S3, the thermograms need to be numbered

·         Figure S11/12/13/14/15: thermograms need to be numbered. Some graphs need to be of higher quality

·         Page 14: be careful it’s “figure S14” and not S13

·         Page 15: it’s “figure S15” and not S14

Round 2

Reviewer 1 Report

no comments at revised version, remarks were adequately answered and/or resulted in changes in the manuscript

Author Response

We appreciate the reviewer's time in rereading our manuscript. 

Reviewer 2 Report

The authors tried to address at their best to the different comments.

The english langage was improved, which is a good point for the reader.

The title was corrected to be more in adequation with the paper content. Yes the pre-formulations studies are required for pharmaceutical development but no it isn't always a separate paper. Some authors combine the pre-formulation studies with the development part to improve the impact of their research and to have a complete paper.

For the hydrogel part, the authors need to be careful because all the hydrogels aren't cross-linked (even the polymeric one), and especially the physical hydrogels. Physical or chemical triggers can be used to induce the crosslinking but only in the case of chemical hydrogels. Semantics is important in this area.

Line 246: the authors need to add the units (2.58 +/- 0.1 and 2.97 +/- 0.09)

Line 384: “ranged from … to…” and not “ranged from … and …”

Line 387: be careful the beginning of this phrase is redundant with line 383/384 (just the previous paragraph). Is it pertinent? It seems that the authors take some part from the conclusion to put it in the R&D part without checking if it was coherent, without redundancy, with the previous paragraph. Thus “decomposition temperature is correct but “temperatures decomposition” isn’t (line 383)

Line 477: Isn’t it “beginning” and not “being” ?

Even if the authors didn't think it could be pertinent to upgrade, I maintain hat some figures aren't of sufficient quality for a publication in the journal
